# Long-term effect of temporary ART initiated during primary HIV-1 infection on viral persistence

Alexander O. Pasternak [1,9] ✉, Pien M. van Paassen[2,9], Yara L. Verschoor [1], Jelmer Vroom[1], Karel A. van Dort[2], Irma Maurer[2], Marlous L. Grijsen [3,4], Ferdinand W. Wit [5,6,7,8], Godelieve J. de Bree [5], Neeltje A. Kootstra [2], Jan M. Prins[5] & Ben Berkhout [1]

Initiation of antiretroviral therapy (ART) during primary HIV-1 infection (PHI) has been proposed to limit the formation of HIV-1 reservoirs. However, it remains unknown whether temporary ART initiated during PHI has a long-term effect on viral persistence. Here, we longitudinally quantify HIV-1 persistence markers and immunological parameters in the participants ($n = 64$) of a randomized controlled trial comparing 24 or 60 weeks of temporary ART vs. no treatment during PHI, who subsequently (re)initiated ART during chronic HIV-1 infection (CHI) after a median period of 116 weeks without treatment (ISRCTN59497461). Levels of several HIV-1 persistence markers (cell-associated unspliced RNA, total DNA, and intact DNA) do not significantly differ and strongly positively correlate between early and CHI ART periods in the same participants. Early ART is associated with lower HIV-1 proviral sequence diversity and superior restoration of the CD4/CD8 ratio, as well as lower levels of monocyte activation markers, compared to CHI ART, in the same participants. At CHI ART, intact HIV-1 DNA negatively correlates with HIV-specific T-cell responses. Finally, levels of HIV-1 persistence markers during CHI ART are lower in participants who had been pre-treated during PHI, indicating a long-term suppressive effect of temporary early ART on the persistence of HIV-1 reservoir.

Combination antiretroviral therapy (ART) suppresses human immunodeficiency virus type 1 (HIV-1) replication, restores immune function, and prevents the development of AIDS[1]. However, ART is not curative and has to be continued lifelong because therapy interruption (TI) almost inevitably results in a fast viral rebound. Persistence of viral reservoirs forms the major obstacle to achieving HIV-1 eradication or a long-term remission[2,3]. In recent years, much effort has been invested in the development of efficient therapeutic strategies to target the reservoirs. In the absence of reliable predictive markers of post-treatment virological control, every such potential curative

[1]Laboratory of Experimental Virology, Department of Medical Microbiology, Amsterdam UMC, University of Amsterdam, Amsterdam, Netherlands. [2]Department of Experimental Immunology, Amsterdam UMC, University of Amsterdam, Amsterdam, Netherlands. [3]Oxford University Clinical Research Unit Indonesia, Faculty of Medicine Universitas Indonesia, Jakarta, Indonesia. [4]Centre for Tropical Medicine and Global Health, Nuffield Department of Medicine, University of Oxford, Oxford, United Kingdom. [5]Department of Internal Medicine, Amsterdam UMC, University of Amsterdam, Amsterdam, Netherlands. [6]Amsterdam Institute for Global Health and Development, Amsterdam, Netherlands. [7]Department of Global Health, Amsterdam Institute for Infection and Immunity, Amsterdam UMC, University of Amsterdam, Amsterdam, Netherlands. [8]HIV Monitoring Foundation, Amsterdam, Netherlands. [9]These authors contributed equally: Alexander O. Pasternak, Pien M. van Paassen. ✉e-mail: a.o.pasternak@amsterdammumc.nl

intervention ultimately necessitates an analytical TI to assess its efficacy. However, the impact of TI on immunological and virological parameters in people with HIV-1 (PWH) has not been fully established, which complicates clinical decision-making and hinders HIV-1 cure research.

Although early ART initiation cannot on itself cure HIV-1 infection[4,5], PWH who initiate ART during acute or early infection achieve lower reservoir levels than those who start therapy during chronic infection[6–16]. Early ART preserves immune function and limits the possibilities for HIV-1 to escape from the host immune response[17], providing a likely explanation for the smaller reservoir under early therapy. In line with this, "post-treatment controllers", PWH who are able to control HIV-1 replication post-TI for extended periods in the absence of ART, are more frequent among those who have initiated ART during early infection[18,19]. Even when it does not lead to post-treatment control, temporary ART initiated during primary HIV-1 infection (PHI) has been shown to delay the viral rebound, lower the plasma viral load set point and defer the restart of ART during chronic HIV-1 infection (CHI)[20–26]. However, so far no study has investigated the long-term effects of temporary ART initiated during PHI on the viral reservoir in PWH who restart ART during CHI.

Here, we measured HIV-1 cell-associated (CA) unspliced (US) RNA, as well as total, intact, and defective HIV-1 DNA, in peripheral blood mononuclear cells (PBMCs) of participants of the Primo-SHM study, a randomized controlled trial (RCT) comparing no treatment with 24 or 60 weeks of temporary ART initiated during PHI[20,27]. All participants who interrupted early ART, or were randomized to the no-treatment arm, (re)initiated ART during CHI after (at median) more than two years without treatment, providing us with the unique opportunity to longitudinally quantify the parameters during both therapy periods. Remarkably, in the participants who had been pre-exposed to ART during PHI, we did not observe any significant differences in the HIV-1 persistence markers between early ART and CHI ART periods. Moreover, levels of US RNA and total DNA during CHI ART were lower in those who had been pre-treated during PHI compared to the no-

treatment arm. Our results indicate that temporary early ART has a long-term suppressive effect on the viral persistence, as revealed during therapy reinitiated after several years.

## Results

### HIV-1 persistence and immunological response dynamics during early ART

All 64 participants of the Primo-SHM study who completed the trial in the Academic Medical Center of the University of Amsterdam (AMC) were included in this study. Baseline and treatment characteristics of the study participants are shown in Table 1 and extensive characteristics are published elsewhere[27]. Participants received no treatment (n = 12), 24 weeks (n = 23), or 60 weeks (n = 29) of early ART (Fig. 1). We started with the longitudinal quantification of US RNA and total DNA during early ART in participants who were randomized to receive 24 or 60 weeks of early ART (n = 52). Parameters were measured at baseline (at PHI, before early ART initiation) and every 12 weeks on ART until TI. Thus, participants of the 24-week arm were assessed at 12 and 24 weeks of early ART, while participants of the 60-week arm were additionally assessed at 36, 48, and 60 weeks. We also assessed plasma viral loads, CD4 + T-cell counts, CD4/CD8 ratios, and calculated HIV-1 transcription levels per provirus (US RNA/total DNA ratios) at the same time points.

As expected, no differences in the longitudinal dynamics of any parameter between the 24-week and 60-week arms were observed (Fig. S1), therefore these two arms were pooled together for further analysis. Figure 2a shows the levels of the measured parameters at baseline and under early ART, while Fig. 3a shows the longitudinal dynamics of these parameters fitted to a two-phase segmentation model. Fig. S2A shows the significance levels of the differences between the time points for all markers. Plasma viral load demonstrated a two-phase decay with a change point at 12 weeks of early ART. By week 24, 92.3% of participants (and by week 36, all remaining participants) achieved virological suppression in plasma to <50 copies/ml and maintained it throughout the therapy period, apart from occasional isolated "blips" that remained below 150 copies/ml. CD4+ count and CD4/CD8 ratio increased between baseline and week 12 but did not significantly change afterwards (Fig. S3A shows the relative increases from baseline of CD4+ counts and CD4/CD8 ratios). Similar to the plasma viral load, US RNA dramatically decreased between baseline and week 12 (median decrease (interquartile range, IQR), 49 (17–389) fold) and remained relatively stable afterwards, although some fluctuations were observed (US RNA levels at 36 weeks were lower than at 24 and 48 weeks ART). In contrast, total DNA showed a continuous decrease until 36 weeks, after which it remained stable. The decay of total DNA during the first phase was much slower than that of US RNA or plasma viral load ($T_{1/2}$ = 12.9 vs. 2.11 vs. 1.17 weeks, respectively). Similar differences in the decay of total HIV-1 DNA and US RNA upon ART initiation were reported by us and others previously[8,28,29]. The gradual decrease in total DNA reflects the elimination of rare cells with intact proviruses by the host immunity in the presence of the large background of defective proviruses that show a much slower decay on ART[30–32], whereas the biphasic kinetics of US RNA reflects the steep decline of productively infected cells upon ART initiation, followed by a quasi-steady state that is fuelled by stochastic reactivation of latently infected cells. Interestingly, the US RNA/total DNA ratio demonstrated a sharp decrease between baseline and week 12 ($T_{1/2}$ = 2.69 weeks), followed by a significant, albeit much slower, increase ($T_d$ = 16.8 weeks).

To assess the possible influence of the ART regimen on the plasma viral load, HIV persistence markers, and immune restoration, we stratified the participants according to the ART class (Fig. S4A) or nucleoside analogue reverse transcriptase inhibitor (NRTI) backbone (Fig. S4B) they received at every time point of early ART. No differences for any marker have been observed between different NRTI

**Table 1 | Baseline and treatment characteristics of the study participants (early ART)**

| | | |
|---|---|---|
| Age, years (n = 64) | | 39.8 (31.7–47.1)[a] |
| Male gender (n = 64) | | 95.3 (61/64) |
| Baseline plasma viral load, log₁₀ copies/ml (n = 64) | | 5.26 (4.63–5.77) |
| Baseline CD4+ count, cells/mm³ (n = 64) | | 490 (310–633) |
| Baseline CD4/CD8 ratio (n = 64) | | 0.46 (0.25–0.79) |
| Time treated with early ART, weeks (n = 64) | 24 | 35.9 (23/64) |
| | 60 | 45.3 (29/64) |
| | No early ART | 18.8 (12/64) |
| NRTI backbone at start of early ART (n = 52) | AZT + 3TC | 63.5 (33/52) |
| | FTC + TDF | 36.5 (19/52) |
| ART class at start of early ART (n = 52) | NNRTI + PI-based | 88.5 (46/52) |
| | PI-based | 9.6 (5/52) |
| | NNRTI-based | 1.9 (1/52) |
| NRTI backbone at interruption of early ART (n = 51)[b] | FTC + TDF | 52.9 (27/51) |
| | AZT + 3TC | 31.4 (16/51) |
| | 3TC + TDF | 15.7 (8/51) |
| ART class at interruption of early ART (n = 51) | NNRTI-based | 64.7 (33/51) |
| | PI-based | 25.5 (13/51) |
| | NNRTI + PI-based | 9.8 (5/51) |

[a]Data are medians (interquartile ranges) for continuous variables and % (proportions) for discrete variables, unless indicated otherwise.
[b]One participant did not interrupt early ART.

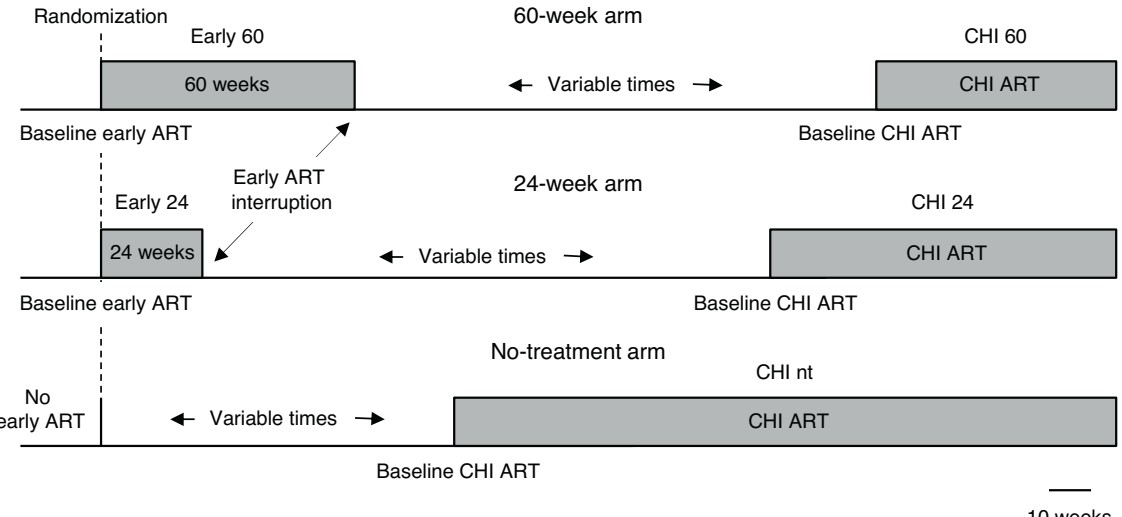

**Fig. 1 | Schematic of the study.** Participants received either no treatment (*n* = 12), 24 weeks (*n* = 23), or 60 weeks (*n* = 29) of early ART. After a period without treatment, participants (re)started ART at chronic HIV infection (CHI). Durations of periods with and without treatment are drawn to scale, as depicted below. The durations of periods without treatment shown in the figure correspond to the median periods without treatment in each of the three arms. nt, no treatment during early ART.

## Table 2 | Baseline and treatment characteristics of the study participants (CHI ART)

| | | |
|---|---|---|
| Baseline plasma viral load, log$_{10}$ copies/ml (*n* = 63)[a] | | 4.67 (4.32–5.21)[b] |
| Baseline CD4+ count, cells/mm³ (*n* = 63) | | 310 (240–410) |
| Baseline CD4/CD8 ratio (*n* = 63) | | 0.36 (0.26–0.43) |
| Time untreated between interruption of early ART or randomization and start of CHI ART, weeks | All participants (*n* = 63) | 115.6 (50.6–200.6) |
| | 24 weeks early ART (*n* = 23) | 134.3 (93.7–231.1) |
| | 60 weeks early ART (*n* = 28) | 123.4 (51.0–199.5) |
| | No early ART (*n* = 12) | 83.4 (33.2–113.4) |
| NRTI backbone at start of CHI ART (*n* = 62)[c] | AZT + 3TC | 3.2 (2/62) |
| | 3TC + TDF | 3.2 (2/62) |
| | FTC + TDF | 93.5 (58/62) |
| ART class at start of CHI ART (*n* = 62) | NNRTI-based | 58.1 (36/62) |
| | PI-based | 32.3 (20/62) |
| | INSTI-based | 4.8 (3/62) |
| | NNRTI + PI-based | 4.8 (3/62) |
| NRTI backbone at 96 weeks of CHI ART (*n* = 61)[d] | ABC + 3TC | 3.3 (2/61) |
| | AZT + 3TC | 1.6 (1/61) |
| | FTC + TDF | 95.1 (58/61) |
| ART class at 96 weeks of CHI ART (*n* = 61) | NNRTI-based | 50.8 (31/61) |
| | PI-based | 37.7 (23/61) |
| | INSTI-based | 11.5 (7/61) |

[a]One out of 64 study participants did not interrupt early ART.
[b]Data are medians (interquartile ranges) for continuous variables and % (proportions) for discrete variables, unless indicated otherwise.
[c]No CHI ART data were available for one participant.
[d]One participant stopped CHI ART before 96 weeks.

backbones. However, plasma viral load and total HIV-1 DNA levels were higher at 12 weeks ART in the participants treated with the triple-class, four-drug ART regimen that included a nonnucleoside reverse transcriptase inhibitor (NNRTI) combined with a protease inhibitor (PI), compared to those treated with the three-drug NNRTI-based regimen

(Fig. S4A). The reason for this difference is that, by the study protocol, almost all participants started the early ART on the four-drug regimen (NNRTI + PI) and were advised to discontinue the fourth drug when their plasma viral load became undetectable[20,27]. Consequently, at 12 weeks ART, most of participants whose plasma viral loads were not yet suppressed (and who, therefore, had higher total HIV-1 DNA levels) were still on the four-drug regimen, while most of those with suppressed viral loads have already discontinued the fourth drug. No other differences by the ART class were observed for any marker.

At baseline and at every time point during early ART, we assessed pairwise correlations between the measured parameters (Fig. S5A). Positive correlations were observed between the virological markers, as well as between the CD4+ count and CD4/CD8 ratio, whereas the virological markers negatively correlated with the CD4+ count and CD4/CD8 ratio. At baseline, correlations were generally stronger than during ART. Both at baseline and under ART, virological markers correlated stronger with the CD4/CD8 ratio than with the CD4+ count. Strong positive correlations were also observed for CD4+ count, CD4/CD8 ratio, and total DNA between the time points, whereas these correlations were weaker for US RNA and especially for the US RNA/total DNA ratio (Fig. S5B). These differences reflect more stable dynamics of total DNA under ART as compared to US RNA, which is known to fluctuate in time[33,34].

### HIV-1 persistence and immunological response dynamics during CHI ART

According to the trial protocol, participants had to (re)start ART in the case of two consecutive CD4+ counts <350 cells/mm³, diagnosis of symptomatic HIV-1 disease, or insistence on (re)starting ART by the physician or participant. 63 out of 64 study participants (re)started ART during CHI (one participant from the 60-week arm chose not to interrupt early ART). Baseline and CHI ART characteristics are shown in Table 2. Median times without treatment between the interruption of early ART (for the 60-week and 24-week arms) or randomization (for the no-treatment arm) and the start of CHI ART were 115.6 (IQR, 50.6–200.6) weeks for all participants, 134.3 (93.7–231.1) weeks for the 24-week arm, 123.4 (51.0–199.5) weeks for the 60-week arm, and 83.4 (33.2–113.4) weeks for the no-treatment arm. The time without treatment was significantly shorter for the no-treatment arm compared to the 24-week arm (Fig. S6), confirming the earlier results of

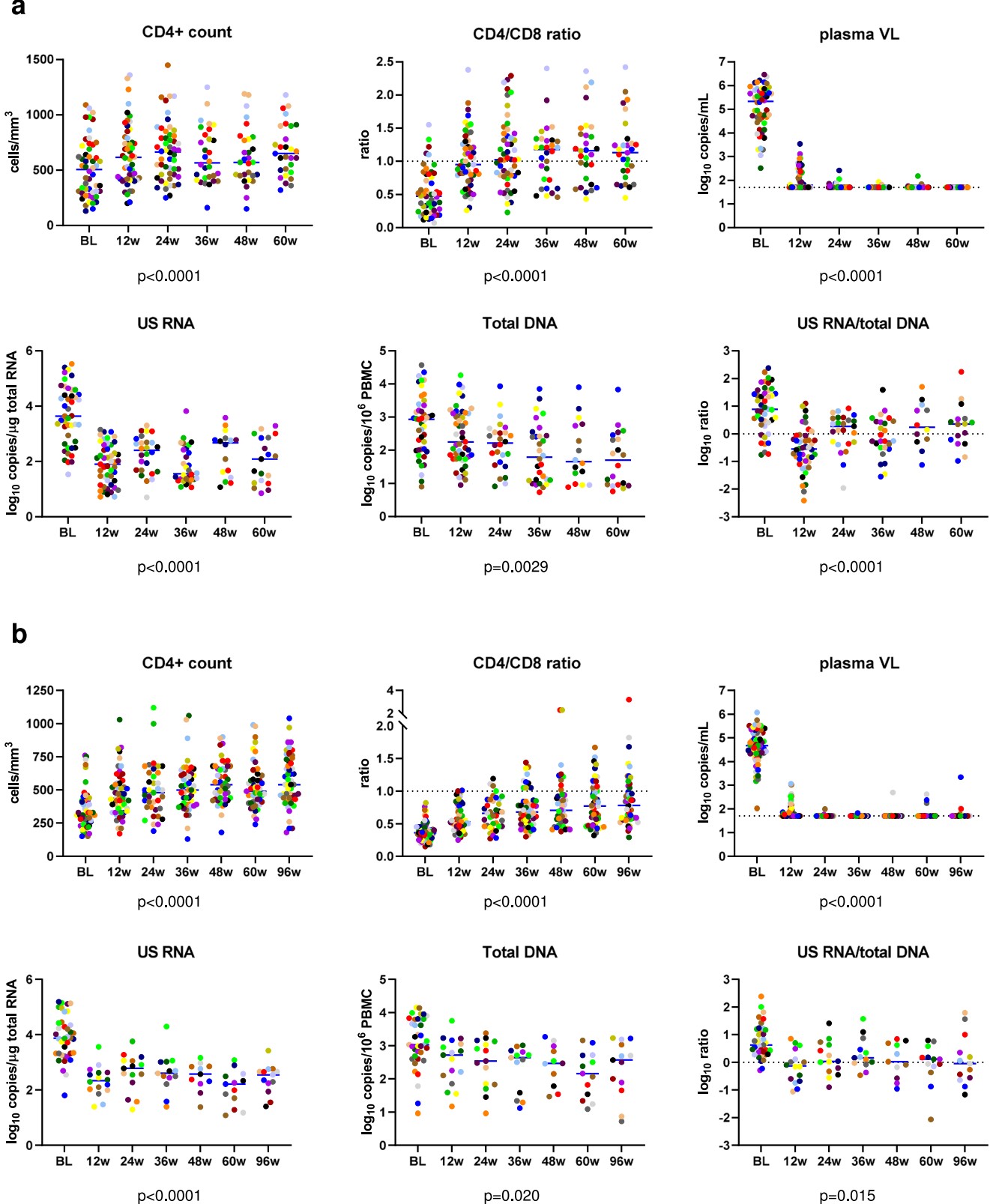

**Fig. 2 | Levels of HIV-1 persistence and immunological response parameters.** Levels of parameters at baseline (BL) and at 12, 24, 36, 48, 60 weeks of early ART (**a**) and 12, 24, 36, 48, 60, and 96 weeks CHI ART (**b**) are shown. Participants are color-coded. For plasma viral load (VL), the limit of detection of the commercial assays (50 copies/mL) is shown with a dashed line. Data points represent individual participants ($n = 52$ for early ART (all parameters), $n = 64$ for CHI ART (CD4+ count, CD4/CD8 ratio, plasma VL), $n = 39$ for CHI ART (US RNA, total DNA, US RNA/total DNA ratio)). Repeated-measures mixed-effects analyses with Tukey corrections for multiple comparisons were used for data analysis. *P* values represent the significance of the change of each parameter between the time points. Source data are provided as a Source Data file.

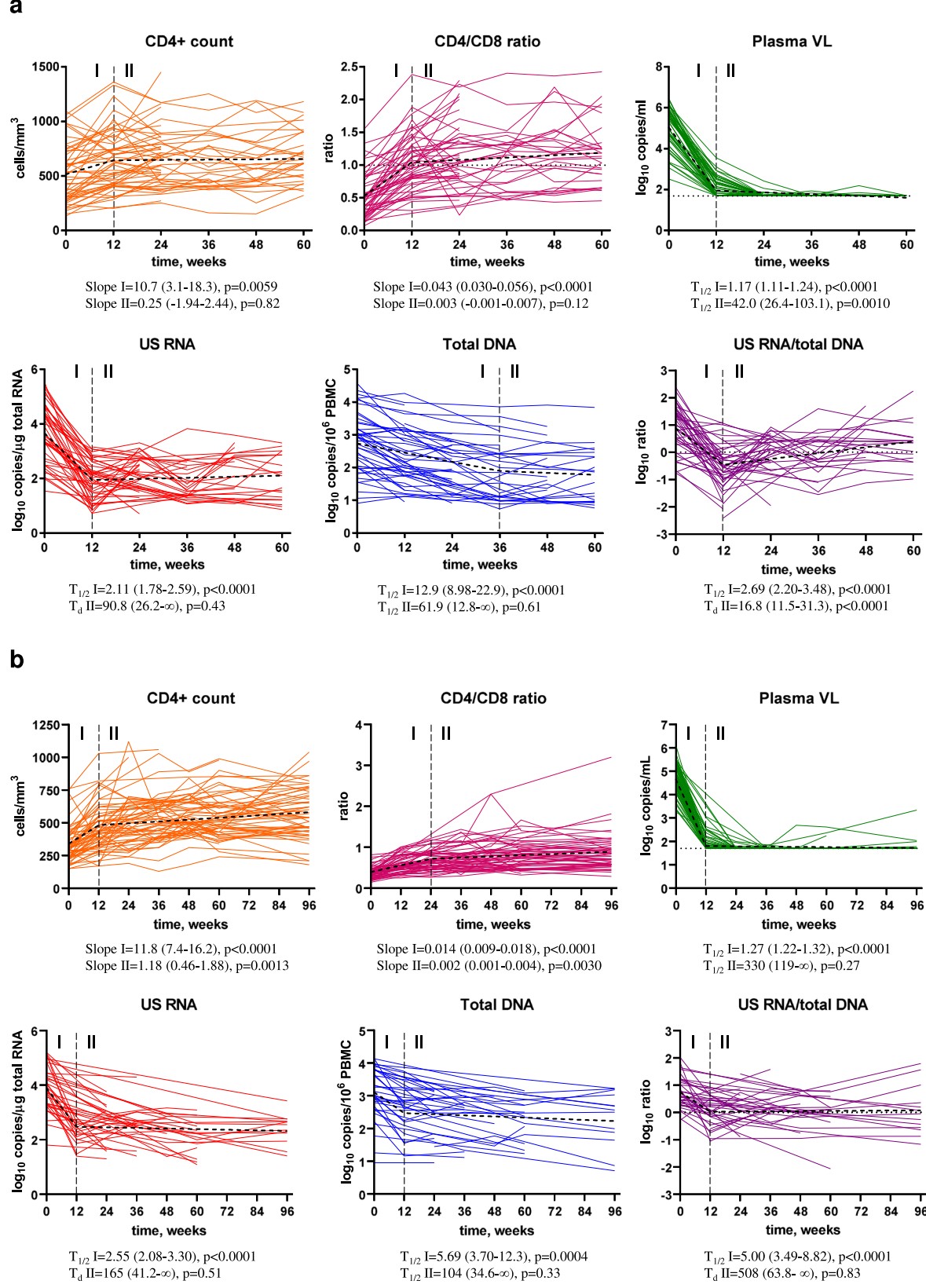

**Fig. 3 | Longitudinal dynamics of HIV-1 persistence and immunological response.** Longitudinal dynamics of measured parameters during early (**a**) and CHI (**b**) ART were fitted to a two-phase segmentation model with change points chosen for every parameter based on the optimal model fit. For CD4+ count and CD4/CD8 ratio, slope values and their 95% confidence intervals during phases I and II are shown. For log-transformed virological biomarkers, half-lives ($T_{1/2}$) or doubling times ($T_d$) and their 95% confidence intervals during phases I and II are shown. Extra sum of squares $F$ tests were used to analyse the data. $P$ values indicate the significance of longitudinal changes in the measured parameters during phases I and II. Source data are provided as a Source Data file.

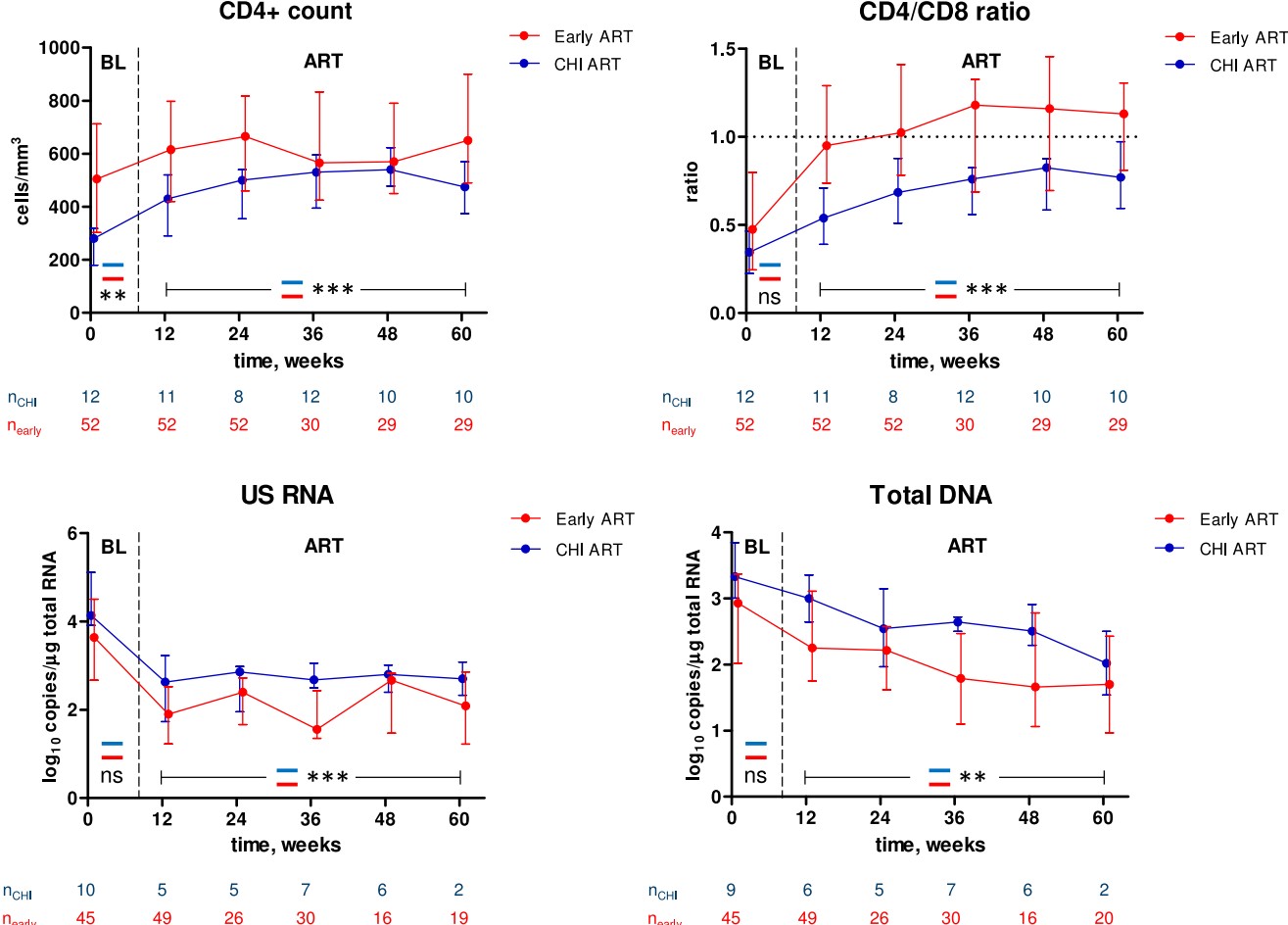

**Fig. 4 | Comparisons between early and CHI ART in different participants.** Parameters were compared between participants treated with early ART (red) and the no-treatment arm during the first 60 weeks of CHI ART (blue). Median values and interquartile ranges are shown. Baseline parameters were compared using Mann–Whitney tests (two-sided) and parameters measured under ART were compared using repeated-measures mixed-effects modelling. Numbers of participants per time point are indicated below the graphs. ***, $p < 0.001$; **, $0.001 < p < 0.01$; ns, not significant. Exact $p$ values are as follows. CD4 count: $p = 0.0035$ (baseline), $p = 1.3*10^{-5}$ (ART). CD4/CD8 ratio: $p = 0.15$ (baseline), $p = 6.6*10^{-8}$ (ART). US RNA: $p = 0.10$ (baseline), $p = 2.1*10^{-5}$ (ART). Total DNA: $p = 0.051$ (baseline), $p = 0.0058$ (ART). Source data are provided as a Source Data file.

Grijsen et al.[20]. During CHI ART, the vast majority of participants were receiving tenofovir/emtricitabine as an NRTI backbone and an NNRTI or a ritonavir-boosted PI as a third drug, although a small minority were receiving an integrase strand transfer inhibitor (INSTI)-based regimen (Table 2). No differences by the ART class were observed for any marker during CHI ART (Fig. S7A) and neither NRTI backbone nor ART class was significantly different between the study arms (Fig. S7B).

We longitudinally quantified the same virological and immunological parameters during CHI ART as described above for the early ART period. Plasma viral loads, CD4+ counts, and CD4/CD8 ratios were measured in all 63 participants treated with CHI ART. US RNA and total DNA were measured in 39 participants, for whom longitudinal PBMC samples were available (16 participants from the 60-week arm, 12 from the 24-week arm, and 11 from the no-treatment arm). Figure 2b shows the levels of the measured parameters at baseline (the time point before CHI ART initiation) and during the first 96 weeks of CHI ART, while Fig. 3b shows the longitudinal dynamics of these parameters fitted to a two-phase segmentation model. Fig. S2B shows the significance levels of the differences between the time points for all markers. In general, the longitudinal dynamics were similar to that observed during early ART, with all parameters fitting to a two-phase model. However, some differences were observed. In contrast to the early ART, where both CD4+ count and CD4/CD8 ratio remained stable after the initial increase, at CHI ART both these parameters continued to increase during the second phase as well, albeit much more slowly than during the first phase. Fig. S3B shows the relative increases from baseline of CD4+ counts and CD4/CD8 ratios during CHI ART. At CHI ART, the US RNA/total DNA ratio remained stable after the initial decrease, whereas an increase in this ratio was observed during the second phase at the early ART.

We then assessed the longitudinal decay dynamics of HIV-1 markers (plasma viral load, total DNA, and US RNA) by comparing their log-transformed relative changes from baseline at early and CHI ART. At all time points, the change in total DNA was the smallest, followed by the change in US RNA, while the change in plasma viral load was the most prominent (Fig. S8). Pairwise comparisons between the relative changes of different markers from baseline revealed significant differences, both during early and CHI ART. When visually comparing early and CHI ART, the decay dynamics of total DNA and US RNA were similar, whereas plasma viral load decreased by a larger magnitude on early ART than on CHI ART (Fig. S8). This reflects higher plasma viral loads at PHI (the baseline of early ART) than at CHI (Fig. 2).

Finally, we assessed pairwise correlations between the measured parameters at baseline and at every time point during CHI ART (Fig. S9A). Similarly to the early ART, we observed positive correlations between the virological markers, as well as between the CD4+ count

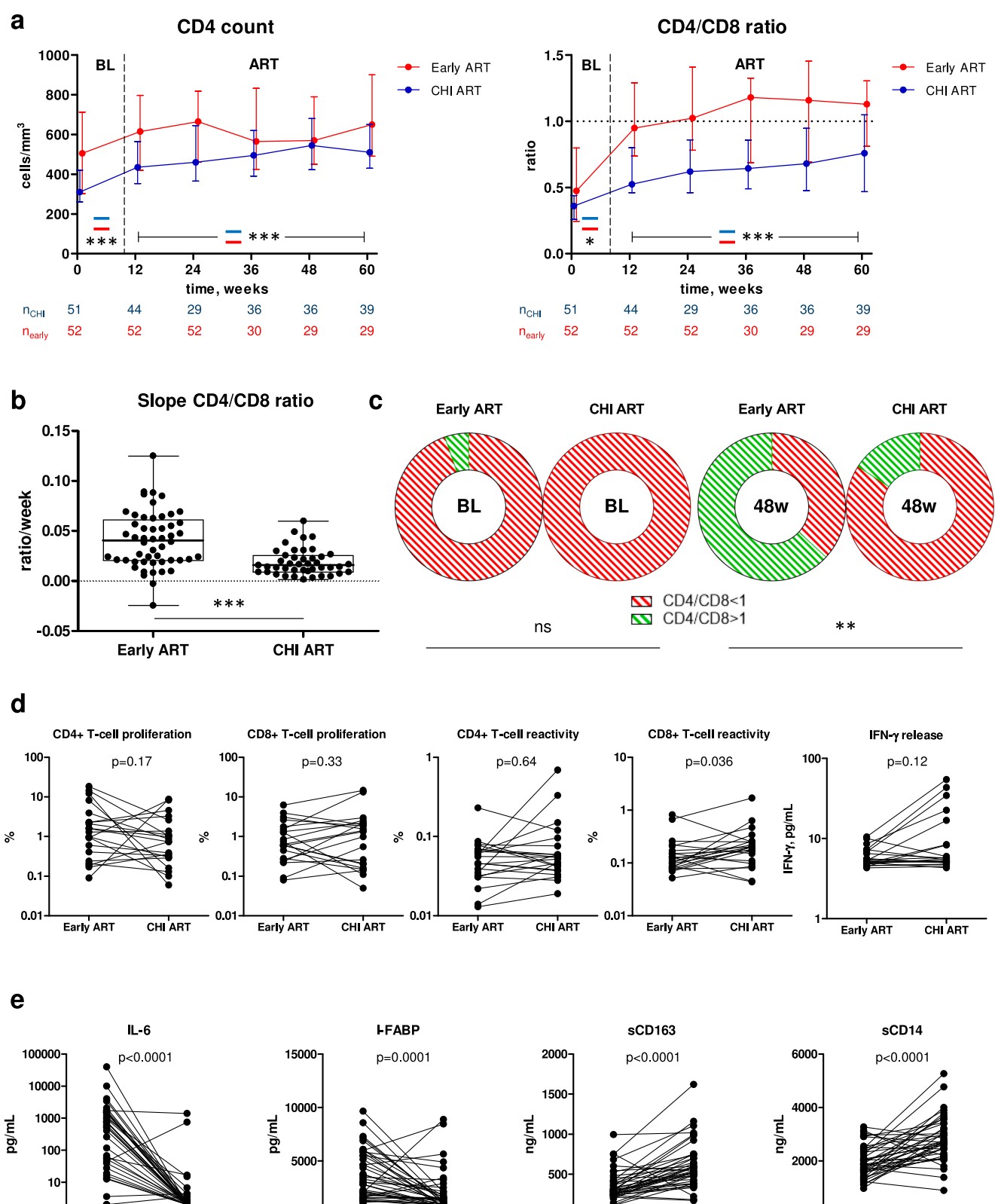

and CD4/CD8 ratio, and negative correlations between virological markers and CD4+ count or CD4/CD8 ratio. However, the correlations at CHI and during CHI ART were weaker than the corresponding correlations at PHI and during early ART (Figs. S5A and S9A). As for early ART, positive correlations were observed for CD4+ count, CD4/CD8 ratio, and total DNA between the time points, whereas these correlations were weaker for US RNA and the US RNA/total DNA ratio (Fig. S9B).

**Fig. 5 | Comparisons of immunological response to early and CHI ART in the same participants. a** Comparisons of CD4+ count and CD4/CD8 ratio between early (red) and CHI ART (blue) periods. Median values and interquartile ranges are shown. Baseline parameters were compared using Mann–Whitney tests (two-sided) and parameters measured under ART were compared using repeated-measures mixed-effects modelling. Numbers of participants per time point are indicated below the graphs. *, 0.01 <p < 0.05; ***, p < 0.001. Exact p values are as follows. CD4 count: p < 0.0001 (baseline), p = 7.5*10$^{-8}$ (ART). CD4/CD8 ratio: p = 0.035 (baseline), p = 4.4*10$^{-19}$ (ART). **b** Comparison of the CD4/CD8 ratio slopes during the first 12 weeks of early and CHI ART in the same participants. Box-and-whiskers plots show the median, quartiles, minimum and maximum values. Data points represent individual participants (n = 52 for early ART, n = 43 for CHI ART). Wilcoxon signed-rank test (two-sided) was used for the comparison. ***, p < 0.0001. **c** Normalization of CD4/CD8 ratio between baseline (BL) and 48 weeks of early and CHI ART in the

same participants. Proportions of participants with CD4/CD8 ratio <1 (red) and >1 (green) are depicted with doughnut charts and compared between early and CHI ART by Fisher's exact tests (two-sided). **, 0.001<p < 0.01; ns, not significant. **d** Comparisons of functional HIV-specific CD4+ and CD8 + T-cell responses between early and CHI ART in the same participants. Data points represent individual participants (n = 21 for CD4+ and CD8 + T-cell proliferation, n = 22 for CD4+ and CD8 + T-cell reactivity and IFN-γ release). Wilcoxon signed rank tests (two-sided) were used to calculate statistical significance. **e** Comparisons of plasma biomarkers of systemic inflammation, intestinal damage, and monocyte activation between early and CHI ART in the same participants. Data points represent individual participants (n = 41). Wilcoxon signed rank tests (two-sided) were used to calculate statistical significance. For all panels, source data are provided as a Source Data file.

## Comparisons between early and CHI ART in different participants

We then proceeded to compare the CD4+ count, CD4/CD8 ratio, and HIV-1 persistence markers between early and CHI ART. First, we compared these parameters in different participants: parameters measured during early ART were compared with the corresponding parameters measured during the first 60 weeks of CHI ART in the no-treatment arm (Fig. 4). Separate comparisons were performed for baseline (PHI vs. CHI) and on-ART values. For the latter comparisons, we used mixed-effects modelling to account for the correlations of longitudinal measurements within participants. The CD4+ count was significantly higher both at PHI than at CHI and during early ART than during CHI ART. In contrast, no difference between PHI and CHI was observed for the CD4/CD8 ratio at baseline but this ratio was significantly higher during early ART than during CHI ART. Accordingly, no difference between early and CHI ART was observed in the relative increases of the CD4+ count from baseline, but for the CD4/CD8 ratio this difference was significant, with a more prominent increase during early ART (Fig. S10A). For both total DNA and US RNA, we observed nonsignificant trends towards lower values at PHI compared to CHI at baseline, and both these markers were significantly lower during early than during CHI ART (Fig. 4).

## Comparisons of immunological response to early and CHI ART in the same participants

Next, we compared the CD4+ count and CD4/CD8 ratio during early and subsequent CHI ART periods in the same participants. As above, the CD4+ count was higher both at PHI than at CHI and during early ART than during CHI ART (Fig. 5a). The CD4/CD8 ratio was slightly higher at PHI than at CHI, but the difference between CD4/CD8 ratios during early and CHI ART periods was much larger than at baseline (Fig. 5a). Relative increases of the CD4+ count and CD4/CD8 ratio from baseline confirmed this, with a significant difference between early and CHI ART for the CD4/CD8 ratio but not for the CD4+ count (Fig. S10B). The difference in the rates of CD4/CD8 ratio restoration was the most prominent during the first 12 weeks of ART, as the CD4/CD8 ratio slopes during early ART were significantly steeper than those during CHI ART in the same participants in the first 12 weeks of ART (Fig. 5b) but not thereafter (Table S1).

We then compared the immunological response to early and CHI ART in the same participants. Normalization of the CD4/CD8 ratio to more than 1 is considered an important measure of immunological response to ART, and a low CD4/CD8 ratio is a prognostic marker for both opportunistic infections and non-AIDS morbidity and mortality[35,36]. Therefore, we compared the normalization of the CD4/CD8 ratio in the first year of early and CHI ART in the same 19 participants, for whom the CD4/CD8 ratio measurements at baseline and 48 weeks of both early and CHI ART were available (only the 60-week arm was included in this analysis). No difference between PHI and CHI was observed in the proportions of participants with the normalized CD4/

CD8 ratio at baseline, as this ratio was <1 in 18 of 19 participants at PHI and all 19 participants at CHI (Fig. 5c). However, by week 48 of early ART, 12 of 19 participants (63%) achieved the CD4/CD8 ratio of >1, whereas this threshold was achieved by only 3 of the same 19 participants (16%) by week 48 of CHI ART (p = 0.0069) (Fig. 5c). Thus, in the same PWH, early ART was superior to the subsequent CHI ART in restoration of the CD4/CD8 ratio.

We then compared functional HIV-specific T-cell responses between early and CHI ART periods in the same participants. To this end, we selected 22 participants, for whom paired PBMC samples obtained at early and CHI ART with similar times from ART initiation were available (median (IQR) difference in times on ART, 4.9 (1.8–11.6) weeks) and performed cross-sectional pairwise comparisons of CD4+ and CD8 + T-cell proliferative responses, reactivity (activation-induced marker (AIM) assay), and IFN-γ release upon stimulation with Gag peptide pools. Robust CD4+ and CD8 + T-cell responses were observed at both early and CHI ART. A significant difference between the ART periods was observed for Gag-specific CD8 + T-cell reactivity that was higher during CHI ART, accompanied by a nonsignificant trend towards higher IFN-γ release during CHI ART (Fig. 5d).

Finally, we compared levels of plasma soluble markers of systemic inflammation (IL-6), intestinal damage (I-FABP), and monocyte activation (sCD163 and sCD14) between early and CHI ART periods in 41 participants with paired plasma samples from early and CHI ART periods (median (IQR) difference in times on ART, 6.0 (2.9–11.1) weeks). Levels of IL-6 and I-FABP were significantly higher, while both monocyte activation markers were significantly lower, on early ART compared to CHI ART in the same participants (Fig. 5e).

## Comparisons of HIV-1 persistence markers between early and CHI ART in the same participants

Next, we compared total DNA and US RNA during early and subsequent CHI ART periods in the same participants (n = 28). Remarkably, in contrast to the differences observed above for these markers between early ART and CHI ART in the no-treatment arm, we measured no significant differences for any of these markers in the same participants (Fig. 6a). We then performed a sensitivity analysis, in which, for each time point, we only retained paired early and CHI ART measurements of CD4+ count, CD4/CD8 ratio, total DNA, and US RNA (Fig. S11). This analysis did not change the conclusions as the results were very similar to those shown in Figs. 5a and 6a. Furthermore, we observed strong correlations between the average levels of total DNA or US RNA measured during early and CHI ART (Fig. 6b). These results demonstrate that, upon reinitiating ART during CHI, PWH who have been temporarily pre-treated during PHI achieve similar levels of HIV-1 persistence markers to those achieved during early ART.

We then asked whether HIV-1 sequence diversity differs between early and CHI ART in the same participants. To determine this, we used single-genome sequencing (SGS) to measure the nucleotide diversity of the 1.4 kb p6 – protease (PR) – reverse transcriptase (RT) region[37,38]

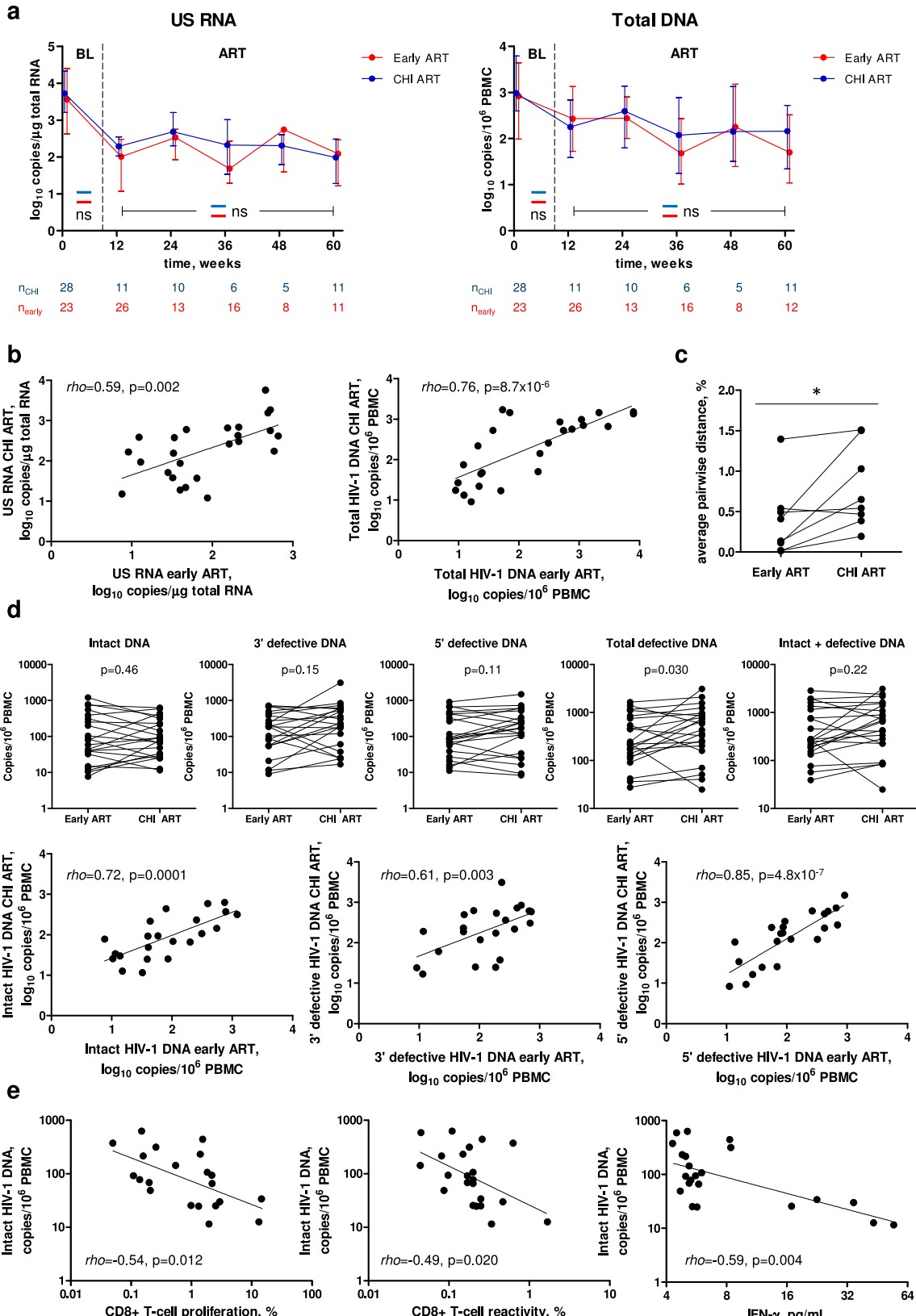

in the total HIV-1 DNA isolated from paired PBMC samples of 8 participants obtained at the same time points during early and CHI ART (12 weeks ART, $n = 6$; 24 weeks ART, $n = 2$). HIV-1 diversity, calculated as average pairwise distance (APD), was significantly higher during CHI ART than during early ART ($p = 0.023$) (Fig. 6c). This difference likely reflects the (on average) 2.5-year period without treatment

between the early and CHI ART, during which HIV-1 nucleotide substitutions are expected to accumulate due to the unsuppressed virus replication[39].

To obtain a deeper insight into possible differences in the persistence of HIV-1 reservoir between early and CHI ART in the same participants, we used intact proviral DNA assay (IPDA) to perform a

**Fig. 6 | Comparisons of HIV-1 persistence markers between early and CHI ART in the same participants. a** Comparisons of US RNA and total HIV-1 DNA between early (red) and CHI ART (blue) periods. Median values and interquartile ranges are shown. Baseline parameters were compared using Mann-Whitney tests (two-sided) and parameters measured under ART were compared using repeated-measures mixed-effects modelling. Numbers of participants per time point are indicated below the graphs. ns, not significant. Exact p values are as follows. US RNA: $p = 0.52$ (baseline), $p = 0.077$ (ART). Total DNA: $p = 0.19$ (baseline), $p = 0.77$ (ART). **b** Correlations between the average levels of US RNA or total DNA measured during early and CHI ART in the same participants. Data points represent individual participants (n = 25). Spearman tests (two-sided) were used to calculate statistical significance. **c** Comparison of HIV-1 sequence diversity between early and CHI ART

in the same participants. HIV-1 diversity was calculated as average pairwise distance (APD). Data points represent individual participants (n = 8). Wilcoxon signed rank test (two-sided) was used to calculate statistical significance. *, $0.01 < p < 0.05$. Exact p value: $p = 0.023$. **d** Comparisons and correlations of intact and defective HIV-1 DNA levels between early and CHI ART in the same participants. Data points represent individual participants (n = 22). Wilcoxon signed rank tests (for comparisons) and Spearman tests (for correlations) were used to calculate statistical significance (all tests were two-sided). **e** Correlations between functional HIV-specific T-cell responses and intact HIV-1 DNA at CHI ART. Data points represent individual participants (n = 21 for CD8 + T-cell proliferation, n = 22 for CD8 + T-cell reactivity and IFN-γ release). Spearman tests (two-sided) were used to calculate statistical significance. For all panels, source data are provided as a Source Data file.

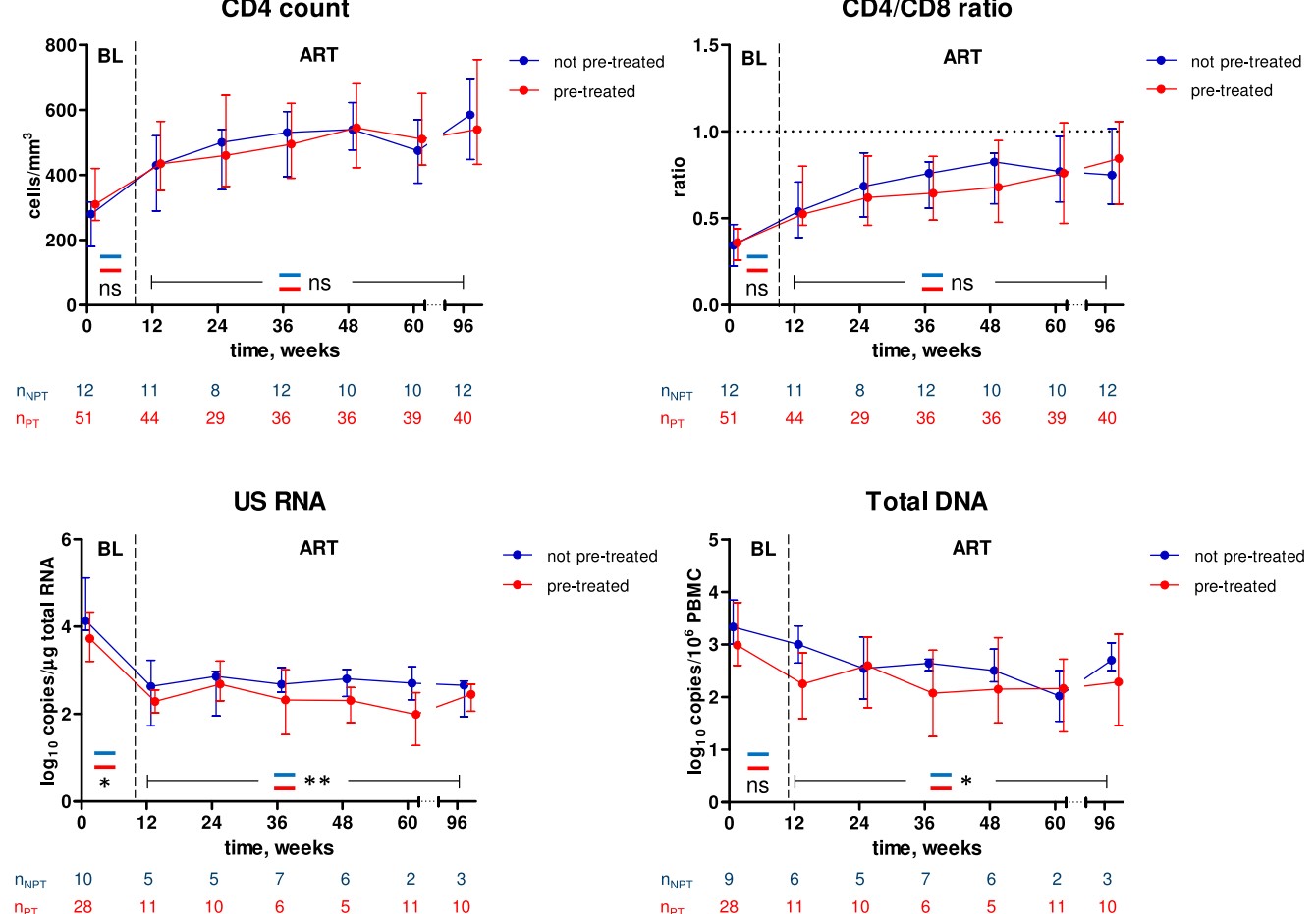

**Fig. 7 | Comparisons between pre-treated and not pre-treated participants during CHI ART.** Parameters were compared between participants who had (red) and had not (blue) been pre-treated with early ART. Median values and interquartile ranges are shown. Baseline parameters were compared using Mann–Whitney tests (two-sided) and parameters measured under ART were compared using repeated-measures mixed-effects modelling. Numbers of participants per time point are

indicated below the graphs. **, $0.001 < p < 0.01$; *, $0.01 < p < 0.05$; ns, not significant. Exact p values are as follows. CD4 count: $p = 0.081$ (baseline), $p = 0.21$ (ART). CD4/CD8 ratio: $p = 0.61$ (baseline), $p = 0.91$ (ART). US RNA: $p = 0.038$ (baseline), $p = 0.0047$ (ART). Total DNA: $p = 0.23$ (baseline), $p = 0.017$ (ART). Source data are provided as a Source Data file.

cross-sectional comparison of levels of intact and defective proviruses in 22 participants with paired samples obtained at early and CHI ART, in whom we also measured T-cell responses (see above). Similarly to what was observed for US RNA and total DNA, no significant difference in the levels of intact proviruses between the therapy periods was measured in the same participants (Fig. 6d). However, nonsignificant trends towards higher levels at CHI ART were observed for both 3' defective and 5' defective proviruses, and significantly higher levels of total defective (3' defective + 5' defective) proviruses were observed at CHI ART (Fig. 6d). In spite of this difference, total (intact + defective)

proviruses did not significantly differ between the two therapy periods (Fig. 6d). We also observed strong positive correlations between the two therapy periods for all proviral forms measured: intact, 3' defective, and 5' defective proviruses (Fig. 6d). Taken together, these results strongly suggest that in these participants, upon reinitiation of ART during CHI, intact HIV-1 reservoirs returned to their early-ART levels despite the long TI. At the same time, an increase in defective proviruses at CHI ART compared to early ART was observed, which may be explained by clonal expansion of cells harbouring defective proviruses[40].

Finally, we determined correlations between the HIV-1 proviral forms and HIV-specific T-cell responses at both early and CHI ART. At early ART, intact HIV-1 DNA weakly negatively correlated with CD8 + T-cell reactivity and IFN-γ release but a correlation of 5′ defective HIV-1 DNA with IFN-γ release was stronger (Fig. S12). However, at CHI ART, intact HIV-1 DNA significantly negatively correlated with CD8 + T-cell proliferation, reactivity, and IFN-γ release (Fig. 6e), while correlations of defective HIV-1 DNA with T-cell responses were weaker (Fig. S12). These results suggest that HIV-specific T-cell responses may restrict the intact HIV-1 reservoir during ART, especially during ART initiated at CHI.

## Comparisons between pre-treated and not pre-treated participants during CHI ART

Taken together, the above results suggested that temporary early ART had a suppressive effect on the HIV-1 persistence markers but did not affect the CD4+ count or CD4/CD8 ratio measured during ART restarted during CHI. To prove this, we compared the parameters during the first 96 weeks of CHI ART between participants who were or were not pre-treated with early ART. Indeed, while no difference was observed for the CD4+ count or CD4/CD8 ratio, both US RNA and total DNA were significantly lower during CHI ART in those participants who had been pre-treated with temporary early ART (Fig. 7). This indicates that temporary early ART had a long-term suppressive effect on the viral persistence, as revealed during therapy reinitiated after several years. We also observed lower levels of US RNA (but not total DNA) at the CHI ART baseline in the pre-treated participants (Fig. 7). These results are in line with the earlier report of Grijsen et al., who measured a lower plasma viral load set point in those Primo-SHM study participants who were treated with early ART[20]. Taken together, this evidence indicates that temporary early ART caused a reduction in the levels of both HIV-1 replication and persistence, observed at CHI and during CHI ART, respectively.

To elucidate any possible long-term effects of the duration of early ART, we compared the parameters during CHI ART between participants who had been pre-treated for 24 weeks, 60 weeks, or not at all, during PHI (Fig. S13). No differences between arms were observed for CD4+ count. The CD4/CD8 ratio during CHI ART was significantly higher in the 24-week arm compared to the 60-week arm but no differences with the no-treatment arm were observed. In contrast, US RNA was significantly lower in both 24-week and 60-week arms, compared to the no-treatment arm, while no difference between 24-week and 60-week arms was observed. Total DNA was significantly lower only in the 60-week arm compared to the no-treatment arm.

Finally, we asked whether the long-term effects of early ART on the viral persistence during CHI ART could be mediated through better antiviral immune responses in pre-treated participants. To this end, we selected 40 participants, for whom PBMC samples at CHI ART were available (60-week arm, n = 16; 24-week arm, n = 14; no-treatment arm, n = 10) and performed cross-sectional comparisons of functional HIV-specific T-cell responses (CD4+ and CD8 + T-cell proliferation and reactivity, as well as IFN-γ release) between the arms. Although it was not possible to perfectly match the participants for times from CHI ART initiation, there were no significant differences in times on ART between the arms (Fig. S14A). Robust CD4+ and CD8+ responses were observed in all arms but no significant differences were observed between the arms for any of the measured HIV-specific T-cell response parameters (Fig. S14B). We also measured soluble markers of systemic inflammation, intestinal damage, and monocyte activation in 53 participants (60-week arm, n = 23; 24-week arm, n = 18; no-treatment arm, n = 12) at CHI ART. No significant differences between the arms were observed for any of these markers as well (Fig. S14C).

## Discussion

In this study, we used longitudinal samples from participants of an RCT who received temporary ART during PHI, underwent a long-term TI, and subsequently restarted ART during CHI. Moreover, we compared HIV-1 persistence markers in these participants with those from the no-treatment arm of this RCT (participants who were randomized not to receive early ART and who started ART during CHI). To eliminate residual confounding, we performed both within-subject comparisons (early vs. CHI ART in the same participants), as well as comparisons between treatment and no-treatment arms of the RCT. This study design allowed us to obtain a number of important and novel insights into the long-term effects of temporary early ART on the HIV-1 persistence.

First, we compared parameters measured at the early ART with the corresponding parameters measured at the CHI ART in the no-treatment arm and measured significantly lower levels of both total HIV-1 DNA and US RNA at early ART. This observation confirms previous reports on lower HIV-1 reservoir levels during early ART[6–8]. However, in contrast to these previous studies that were observational and compared different PWH cohorts (early vs. late treatment), our RCT design allowed to eliminate, or at least drastically reduce, any potential bias.

Second, we could not measure any significant difference in total HIV-1 DNA, intact HIV-1 DNA, or US RNA levels between early and CHI ART within the same participants. This result was unexpected, given the well-known association of early ART with reduced HIV-1 reservoir levels. Moreover, strong correlations between the two therapy periods were observed for US RNA and total DNA, as well as for intact and defective HIV-1 DNA. Taken together, these results indicate that HIV-1 reservoir markers returned to their early-ART levels upon restart of ART during CHI. Several studies have previously demonstrated that short-term (6–12 weeks) analytical TI does not have any permanent effect on the reservoir size as no significant difference in HIV-1 persistence markers pre- vs. post-TI could be measured[41–44]. It should be noted that these studies were cross-sectional (one pre-TI sample was compared with one post-TI sample per participant), therefore variables were not measured at the same time points from ART initiation and re-initiation. In addition, the participants of these earlier studies were not generally treated during PHI. In contrast, our longitudinal study design allowed to compare the dynamics of HIV-1 persistence markers in the same period after initiation of early and CHI ART within the same individuals. Structured TI is an essential component in the design of HIV-1 curative intervention trials. Despite the fact that PWH who restart ART following TI generally resuppress their plasma viral load and recover their CD4+ counts, a TI can still potentially lead to an increase in reservoir. Strikingly, our results demonstrate that if ART is initiated during PHI, then even a long-term (median, 2.5 years) TI does not irreversibly change the reservoir.

Third, we directly demonstrated the long-term effect of temporary early ART on the HIV-1 persistence by comparing viral markers during CHI ART between participants who were randomized to treatment or no treatment during PHI. Both total DNA and US RNA were significantly lower in pre-treated participants, despite that they had been untreated for a longer period during the TI than the no-treatment arm (median, 129 vs. 83 weeks; p = 0.026) and could have accumulated a larger viral reservoir. Interestingly, while 24 weeks of early ART were sufficient to decrease the US RNA levels during CHI, for total DNA this effect was only achieved with 60 weeks of early ART. Further studies should investigate whether a longer period of temporary ART initiated during PHI can result in an even stronger reduction of the reservoir.

To the best of our knowledge, this is the first study to demonstrate long-term effects of temporary early ART on the persistence of HIV-1 reservoir. In the recent years, the reservoir, especially its transcriptionally active part, has been increasingly recognized to persist not only by viral latency but also by resistance to immune clearance of infected cells[2,45–47]. This immune clearance appears to play a surprisingly prominent role in shaping the reservoir by continuously removing the transcription-competent proviruses in a process that has been

termed 'autologous shock and kill'[48]. In line with this, a number of studies have demonstrated HIV-1 US RNA to be a powerful predictor of viral rebound upon ART interruption[27,49–52], suggesting that the transcription-competent reservoir reflects the replication-competent reservoir[53,54]. The mechanisms of HIV-1 immune resistance under ART are under active investigation[55,56], but temporary early ART may either permanently imprint the immune system so that it is more efficient in clearing the reservoir cells during subsequently restarted ART, or suppress the ability of HIV-1 to escape this immune clearance. Either way, our results are in line with earlier evidence from several groups that temporary early ART lowers the plasma viral load set point in the untreated infection after TI[20,21]. In our study, we observed significant negative correlations between intact HIV-1 proviral load and HIV-specific T-cell responses at CHI ART in the pre-treated participants, suggesting the immune system indeed restricts the reservoir during ART. However, we did not observe any differences between the study arms in HIV-specific T-cell responses and soluble markers of systemic inflammation, intestinal damage, and monocyte activation measured during CHI ART. Further studies should unravel the exact mechanisms underlying the effects of temporary early ART on the HIV-1 reservoir.

Despite no difference in HIV-1 persistence marker levels between early and CHI ART in the same participants, proviral diversity was significantly higher during CHI ART than during early ART in the same participants and at the same time points on therapy. In the untreated infection, it has been reported that HIV-1 diversity correlates with viral replicative fitness and diversity before ART initiation predicted delayed viral rebound upon TI[57,58]. This suggests that despite the low reservoir size during CHI ART in the pre-treated PWH, HIV-1 could still be fitter and therefore presumably more prone to rebound if CHI ART were interrupted, compared with the early ART period in these PWH. Indeed, post-treatment control is more frequent after interruption of ART started at PHI[19]. In this study, CD8 + T-cell reactivity and monocyte activation were also higher during CHI ART than during early ART in the same participants, which may reflect a broader antigenic stimulation by a more diverse viral population. We recently demonstrated that levels of sCD14, a monocyte activation marker, measured just before TI, positively correlated with plasma viral load setpoint after TI[59]. Alternatively, the increase in monocyte activation markers at CHI ART could reflect ~2.5 years of untreated infection between early TI and CHI ART initiation. This is consistent with previous studies showing that monocyte activation remains elevated in PWH despite viral suppression[60]. On the other hand, the observed decline in systemic inflammation and intestinal damage markers at CHI ART compared to early ART aligns with previous observations in individuals with CHI[60,61]. The higher levels of inflammatory and gut damage markers during PHI are likely attributable to HIV-induced immune activation in the early stages of HIV-1 infection[62]. This reflects the well-documented immune dysregulation and heightened inflammatory response observed in early HIV-1 infection before ART initiation.

Our study design allowed to compare the immunological response to early and CHI ART in the same PWH. In this respect, a remarkable difference was observed between the two markers of immunological response to ART: CD4+ counts and CD4/CD8 ratios. The participants started early ART at significantly higher CD4+ counts than at the restart of ART during CHI, and this difference was maintained throughout the first 60 weeks of ART. However, early ART did not restore the CD4+ counts faster than CHI ART: in fact, there was no difference between the relative gains in CD4+ counts from baseline between early and CHI ART in the same participants. In contrast, early ART restored the CD4/CD8 ratios significantly faster than CHI ART, with the strongest effect observed during the first 12 weeks of ART. As a result, we observed a large difference in the percentage of participants who had their CD4/CD8 ratios restored to more than 1 by week 48 of early vs. CHI ART, in the same participants. Our results are in agreement with an earlier study that reported greater CD4/CD8 ratio

increases in individuals who initiated ART within six months of infection compared to later initiators[35]. However, an advantage of our current study design is that we compared the CD4/CD8 ratio dynamics on early vs. CHI ART within the same PWH. Because of the independent prognostic value of the CD4/CD8 ratio for non-AIDS morbidity and mortality, our results support the early initiation of ART.

In contrast to its effect on the viral persistence, temporary early ART did not appear to influence the immunological response to CHI ART, as no difference was observed between the pre-treated and not pre-treated arms in either baseline values or reconstitution rates of CD4+ count or CD4/CD8 ratio. These results are in agreement with an earlier report that compared the CD4+ count restoration on CHI ART between the pre-treated and not pre-treated arms in another subset of the Primo-SHM study, and with a report by the Short Pulse Anti-Retroviral Therapy at Seroconversion (SPARTAC) trial investigators[21,63]. However, another study observed that individuals transiently pre-treated during PHI restarted ART with higher baseline CD4+ counts and retained significantly higher CD4+ counts on CHI ART[64]. It should be noted that the latter study included PWH who were pre-treated for different periods. In addition, it was an observational study, therefore residual confounding cannot be entirely excluded.

Our study has some limitations. As the vast majority of participants were MSM residing in the Netherlands who were infected with HIV-1 subtype B[27], the results may not be fully generalizable to other populations differing by sex, transmission route, or HIV-1 subtype. Another limitation is the exclusive focus on peripheral blood. As more than 98% of the body CD4 + T cells are locked in the lymphoid organs, these are the primary sites of HIV-1 replication in untreated PWH. It is unclear, however, whether the infected cell frequency in tissues in ART-treated PWH is higher than in peripheral blood. Further studies should investigate the effects of temporary early ART on HIV-1 persistence in tissues.

In summary, we report here on a long-term suppressive effect of temporary early ART on the persistence of the viral reservoir. Not only have we failed to detect any detrimental effect of transient treatment during PHI on the virological or immunological response to CHI ART, but we even measured lower levels of HIV-1 persistence markers during CHI ART in the pre-treated participants. Moreover, we demonstrate an absence of any detrimental effect of a long TI on HIV-1 persistence. Our findings are helpful for the design of HIV-1 curative interventions that necessitate a treatment interruption.

## Methods
### Study design
The Primo-SHM trial was a multicentre RCT comparing temporary early ART with no treatment during PHI (registration number ISRCTN59497461). Participants were randomly assigned to receive no treatment or 24 or 60 weeks of ART (3-way randomization). If treatment was clinically indicated based on severe clinical symptoms or the participant insisted on starting early ART, subjects were randomized over the 2 treatment arms (2-way randomization). Details of trial design, as well as primary and secondary outcomes of the trial, are reported elsewhere[20]. Of 64 participants included in the present study, 50 (78.1%) were 3-way and 14 (21.9%) were 2-way randomized. Participants were recruited from May 2003 until March 2010, and follow-up data were collected until April 2015. Among participants, 81.3% were diagnosed with HIV-1 during Fiebig stage III or IV of early infection, and treatment started on average 4 weeks after diagnosis[27]. Most participants (82.8%) were MSM, and 84.4% were infected with HIV-1 subtype B[27].

### Ethics statement
The study was approved by the AMC Medical Ethics Committee, and written informed consent was obtained from all participants.

## PBMC and plasma samples

In total, 368 PBMC samples were included in the analysis: 306 for the longitudinal analysis and 62 for the cross-sectional analysis. From the 306 samples used for the longitudinal analysis, 186 samples were from the early ART and 120 from the CHI ART periods. The PHI samples were obtained a median of 2 days (IQR, 0–5 days) before the early ART initiation. For the 12-week measurements on early ART, samples were obtained at a median of 1.1 weeks (IQR, 0.3–2.0 weeks) before or after the exact 12-week time point. For the 24-week, 36-week, 48-week, and 60-week measurements on early ART, these differences were 1.0 (0.0–2.2), 1.0 (0.7–1.5), 1.1 (0.9–1.5), and 0.7 (0.3–1.5) weeks, respectively. The CHI baseline samples were obtained a median of 3.6 (IQR, 0.9–12.8) weeks before the initiation of CHI ART. For the 12-week measurements on CHI ART, samples were obtained at a median of 3.6 weeks (IQR, 2.1–4.9 weeks) before or after the exact 12-week time point. For the 24-week, 36-week, 48-week, 60-week, and 96-week measurements on CHI ART, these differences were 2.8 (1.8–3.6), 2.6 (1.6–4.0), 3.4 (1.9–5.0), 6.7 (3.4–14.6) and 13.6 (8.6–14.6) weeks, respectively. From the 62 PBMC samples used for the cross-sectional analysis, 22 were from the early ART period and 40 from the CHI ART period. We also used 94 plasma samples (41 from the early ART period and 53 from the CHI ART period) to measure soluble markers of systemic inflammation, intestinal damage, and monocyte activation.

## Measurements of virological markers

Plasma viral load was measured using commercial assays with detection limits of 50 or 40 copies/mL. For total HIV-1 DNA and US RNA measurements, total nucleic acids were extracted from PBMCs using the Boom isolation method[65]. Extracted cellular RNA was treated with DNase (DNA-free kit; Thermo Fisher Scientific) to remove DNA that could interfere with the quantitation and reverse-transcribed using random primers and SuperScript III reverse transcriptase (all from Thermo Fisher Scientific). Total HIV-1 DNA and US RNA were measured using previously described qPCR-based methods[66,67]. HIV-1 DNA or RNA copy numbers were determined using a 7-point standard curve with a linear range of more than 5 orders of magnitude that was included in every qPCR run and normalized to the total cellular DNA (by measurement of β-actin DNA) or RNA (by measurement of 18S ribosomal RNA) inputs, respectively, as described previously[29]. Non-template control wells were included in every qPCR run and were consistently negative.

Total HIV-1 DNA was detectable in 97.8% of the PHI samples and in 87.2% of the early ART samples. US RNA was detectable in 95.6% of the PHI samples and in 68.6% of the early ART samples. Both total DNA and US RNA were detectable in 100% of the CHI baseline samples, whereas their detectability in the CHI samples was 97.6% and 90.1%, respectively. Undetectable measurements of US RNA or total DNA were assigned the values corresponding to the corresponding assay detection limits, with a maximum of 200 copies/μg total RNA or 200 copies/$10^6$ PBMCs, respectively. The detection limits depended on the amounts of the normalizer (input cellular DNA or RNA) and therefore differed among samples.

## Measurement of intact and defective HIV-1 proviruses

Intact and defective HIV-1 proviruses were quantified by the IPDA[68]. In brief, genomic DNA was isolated from PBMCs using Puregene Cell Kit (QIAGEN Benelux B.V.) according to the manufacturer's instructions and digested with *Bgl*I restriction enzyme (ThermoFisher Scientific) as described previously[69]. Notably, only a small minority (<8%) of HIV-1 clade B sequences contain *Bgl*I recognition sites between Ψ and *env* amplicons, therefore *Bgl*I digestion is not expected to substantially influence the IPDA output, while improving the assay sensitivity by increasing the genomic DNA input into a droplet digital PCR (ddPCR) reaction[69]. After desalting by ethanol precipitation, genomic DNA was subjected to two separate multiplex ddPCR assays: one targeting HIV

Ψ and *env* regions using primers and probes described previously, including the unlabelled *env* competitor probe to exclude hypermutated sequences[68], and one targeting the cellular *RPP30* gene, which was measured to determine the DNA shearing index[69] and to normalize the intact HIV-1 DNA to the cellular input. The *RPP30* assay amplified two regions, with amplicons located at exactly the same distance from each other as HIV Ψ and *env* amplicons. The first region was amplified using a forward primer 5'-AGATTTGGACCTGCGAGCG-3', a reverse primer 5'-GAGCGGCTGTCTCCACAAGT-3', and a fluorescent probe 5'-FAM-TTCTGACCTGAAGGCTCTGCGCG-BHQ1-3'[70]. The second region was amplified using a forward primer 5'-AGAGAGCAACTTCTTCAAGGG-3', a reverse primer 5'-TCATCTACAAAGTCAGAACATCAGA-3', and a fluorescent probe 5'-HEX-CCCGGCTCTATGATGTTGTTG-CAGT-BHQ1-3'. The ddPCR conditions were as described previously[68] with some minor amendments: we used 46 cycles of denaturation/annealing/extension and the annealing/extension temperature was 60 °C.

## Measurements of HIV-1 sequence diversity

SGS of HIV-1 p6-PR-RT region in the total HIV-1 DNA was performed essentially as previously described[38]. Products of the nested PCR were sequenced directly with BigDye Terminator-based Sanger sequencing using 4 internal primers: 2030+ (5'-TGTTGGAAATGTGGAAAGGAAG-GAC-3'), 2600+ (5'-ATGGCCCAAAAGTTAAACAATGGC-3'), 2610- (5'-TTCTTCTGTCAATGGCCATTGTTTAAC-3'), and 3330- (5'-TTGCCC AATTCAATTTTCCCACTAA-3'). Sequences were assembled using CodonCode Aligner. Chromatograms were examined for double peaks and sequences with evidence of mixed bases were excluded from the analysis. Sequences were also screened for APOBEC-induced G-to-A hypermutation using Hypermut 2.0 software (https://www.hiv.lanl.gov/content/sequence/HYPERMUT/hypermut.html) and the hypermutated sequences were excluded from the analysis. A median of 9 (IQR, 7-11) SGS per sample were included in the final analysis (150 SGS in total). Maximum likelihood analyses and diversity calculations were performed using MEGA 7.0 software. APD was calculated to determine HIV-1 sequence diversity.

## Measurement of functional HIV-specific cellular immune responses

IFN-γ release assay (IGRA) was performed to evaluate the immune response upon HIV-1 peptide pool stimulation. A total of $0.5 \times 10^6$ PBMCs were stimulated with an HIV-1 consensus B Gag peptide pool (1 μg/ml, NIH AIDS Reagent Program) or cultured in medium alone as a control. After 16 hrs, culture supernatants were harvested and IFN-γ released by the cells was determined by human IFN-γ DuoSet ELISA (R&D Systems, Minneapolis, MN, USA).

AIM assay was performed to assess the frequency of reactive CD4+ and CD8 + T cells. PBMCs from the IGRA assay were stained for flow cytometry with FITC CD4 (BD Biosciences), PE OX40, PE-Cy7 CD69, APC Fire-750 CD137, PacBlue CD8 and BV510 CD3 (BioLegend). Reactive T-cells were determined by co-expression of CD137 and OX40 or CD137 and CD69 within the CD4+ and CD8 + T-cell populations, respectively. Fluorescence was measured on the FACS Canto II fluorescence-activated cell sorter (BD Biosciences). Marker expression levels were analyzed using FlowJo version 10.8.1 (TreeStar, Ashland, OR, USA). The gating strategy is shown in Fig. S15.

Proliferation of CD4+ and CD8 + T cells upon antigen stimulation was assessed through the use of the CellTrace™ VioleT-cell Proliferation kit (ThermoFisher). Cells were stained with CellTrace Violet according to manufacturer's protocol (0.5 μM final concentration). Subsequently, the cells were stimulated with an HIV-1 consensus B Gag peptide pool (1 μg/ml final concentration, NIH AIDS Reagent Program). An unstimulated control (medium) and positive control using a peptide pool of CMV pp65 (1 μg/ml final concentration, NIH AIDS Reagent Program) were included. After 6 days, cells were stained with FITC

CD3, PerCP-Cy5.5 CD4 (BD bioscience) and APC CD8 (BioLegend) for 30 minutes at 4 °C. After fixation of the cells with CellFIX (BD), samples were analyzed on the BD FACSCanto™ II to assess the proliferation of CD4+ and CD8 + T cells. The proportion of proliferating cells was determined using FlowJo version 10.8.1 (TreeStar, Ashland, OR, USA). The gating strategy is shown in Fig. S16.

## Measurement of plasma biomarkers
Concentrations of I-FABP, IL-6, sCD14, and sCD163 were determined in plasma samples stored at −80 °C using enzyme-linked immunosorbent assay (ELISA) (I-FABP, IL-6, CD14, and CD163 DuoSet ELISAs; R&D Systems).

## Statistical analysis
Repeated-measures mixed-effects analyses with Tukey corrections for multiple comparisons were used to compare levels of the virological and immunological parameters between time points (at baseline and under ART) and to compare log-transformed relative changes from baseline between the virological parameters. The Greenhouse-Geisser correction was applied to adjust for the lack of sphericity. The longitudinal dynamics of the virological and immunological parameters was fitted to a two-phase segmentation model with change points chosen for every parameter based on the optimal model fit. The slopes were compared with zero, or between therapy periods and study arms, using extra sum of squares F-tests. Repeated-measures mixed-effects analyses and Mann-Whitney tests were used to compare the virological and immunological parameters between different ART classes and NRTI backbones that the participants were receiving during early and CHI ART. Chi-squared tests were used to compare different ART classes and NRTI backbones at baseline and 96 weeks of CHI ART between the study arms. Pairwise correlations between parameters per time point and between time points per parameter were determined by Spearman tests. Times without treatment were compared between the arms using Kruskal–Wallis test with Dunn's multiple comparison posttests. Between-group comparisons of baseline parameters were performed using Mann-Whitney tests. Parameters measured under ART were compared between groups using repeated-measures mixed-effects modelling. Proportions of participants with normalized CD4/CD8 ratios were compared between early and CHI ART by Fisher's exact test. HIV-1 sequence diversity, intact and defective proviral loads, functional HIV-specific T-cell responses, and plasma biomarker levels were compared between early and CHI ART using Wilcoxon signed rank tests. Correlation analyses between early and CHI ART and between virological and immunological parameters were performed using Spearman tests. Functional HIV-specific T-cell responses and plasma biomarker levels were compared between the study arms at CHI ART using Kruskal-Wallis tests. Data were analysed using Prism 10.2.0 (GraphPad Software) and IBM SPSS Statistics (version 28.0.1.0). All tests were two-sided. $P$ values < 0.05 were considered statistically significant.

## Reporting summary
Further information on research design is available in the Nature Portfolio Reporting Summary linked to this article.

## Data availability
Source data are provided for all experimental results presented in the main manuscript and supplementary information. Source data are provided with this paper.

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

## Acknowledgements

We thank Elizabeth Anderson (HIV Dynamics and Replication Program, NIH, Frederick) for assistance with SGS set-up. We are thankful to Margreet Bakker, Luuk Gras, and Colette Smit for their help with data collection. We would like to thank the Primo-SHM study group and study participants for helping establish this cohort. This study was supported by Aidsfonds Netherlands under grant number 2011020 to A.O.P. A.O.P. acknowledges grant support from amfAR, The Foundation for AIDS Research (grant no. 1110680–77-RPRL), and from Partnership NWO-Dutch AIDS Fonds 'HIV cure for everyone' (grant no. KICH2.V4P.AF23.001).

## Author contributions

Conceptualization: A.O.P., M.L.G., F.W.W., G.J.B., N.A.K., J.M.P., B.B. Investigation: A.O.P., P.M.P., Y.L.V., J.V., K.A.D., I.M. Data analysis: A.O.P. Writing—original draft: A.O.P. Writing—review & editing: A.O.P., P.M.P., Y.L.V., M.L.G., F.W.W., G.J.B., N.A.K., B.B. All authors approved the final manuscript.

## Competing interests

The authors declare the following competing interests. F.W.W. has served on the advisory board of ViiV Healthcare, paid to his institution. The other authors declare no competing interests.
