## [Peer Review file · Nature Communications]

Long-term effect of temporary ART initiated during primary HIV-1 infection on viral persistence

Corresponding Author: Dr Alexander Pasternak

Version 0:

Reviewer comments:

Reviewer #1

(Remarks to the Author)

In the study by Pasternak et al, the authors analyzed the long-term suppressive effect of temporary early ART on the persistence of the viral reservoir. The group had access to a group of unique samples derived from a randomized controlled trial in which they compared 24 or 60 weeks of temporary ART vs. no treatment during primary HIV infection (PHI) with data points longitudinally measured after (re) initiation of ART during chronic HIV infection (CHI) after a median period of ~2.5 years without treatment. The first part of the study, in which the authors compared parameters measured at early ART with the counterparts quantified during chronic ART is not novel and it confirms previous results demonstrating lower HIV-1 reservoir levels and better immune profiles if patients are treated with ART early (e.g, PMID: 34841369, PMID: 33105471, etc.).

The novelty of this study stems from the findings that, in pretreated participants, they observed lower levels of viral persistence markers (cell-associated HIV unspliced RNA and total HIV-1 DNA) during the chronic reinitiation phase compared to those subjects that were not treated early. Interestingly, despite having a long treatment interruption, the authors report that they could not find any significant difference in viral persistence parameters between early and chronic ART; this suggests that it may be relatively safe to withdraw ART during future cure interventions.

The main limitation of the manuscript is the confounding effect of different ART regimens administered at the beginning and end of both early and chronic ART phases. Even though the study design likely does not have enough power to discriminate the effect of different ART regimens on viral persistence (usRNA and HIV DNA) or immune responses (CD4 and CD4/CD4 ratio), this potential confounder must be addressed. Tables 1 and 2 reveal that different regimens were administered to the study subjects at different time points. For example, during CHI, some patients but not all received integrase strand transfer inhibitor-based regimens. The authors merely mentioned (lines 146-147), "neither NRTI backbone nor ART class was significantly different between the study arms (data not shown)". Because the switch to certain ART drugs like dolutegravir has been shown to increase the size of the circulation reservoir (e.g, PMID: 38485563), it should be important to stratify patients during both early and chronic ART based on regimen and determine both parameters of viral persistence and immune reconstitution.

A minor limitation is the interpretation of the transcriptional activity of the HIV-1 reservoir. On line 108, the authors claimed that it "remained stable afterward". However, in Figure 2A, they showed statistically significant differences in the levels of US RNA when they compared 24w. vs. 36w and 36w. vs. 48w. They should rephrase these findings.

Reviewer #2

(Remarks to the Author)

Pasternak et al. conducted limited immune and virologic analyses on clinical specimens collected from a previous/follow-up clinical study addressing the impact of early initiation of antiretroviral therapy (ART) on persistent HIV reservoirs and CD4/CD8 counts in people with HIV (PWH). The major findings of this study include: 1) the size of HIV reservoirs is lower in PWH who initiated ART during the early phase of infection, 2) the reservoir size in the early ART group remains relatively low in early treated populations despite having undergone a relatively lengthy treatment interruption, and 3) this is one of the first longitudinal studies addressing the effect of short-term ART in PWH who later became chronic-infected and subsequently reinitiated ART. Although the data presented in this manuscript are somewhat expected, this study is unique in

that the key measurements were done in the longitudinal early-treated study cohort who underwent treatment interruption and ART. However, the current manuscript lacks any explanation regarding the underlying mechanism(s) by which the early initiation of ART allows better immunologic and virologic control in the study populations examined.

1. This manuscript, especially the abstract section, could have benefited tremendously if the authors had clearly defined the study populations and the time points when the assays were conducted. Unless you are intimately familiar with the original study design and read the rest of the manuscript first, it is challenging to comprehend the abstract mainly because the “early” population eventually became “chronic” during treatment interruption and it is not clear what time points immune and virologic markers were measured (i.e., during initial or later ART period). It may not be a bad idea to give each population an acronym (for example, “Early 24” etc). In addition, given the early study participants eventually became chronic before reinstitution of ART, it is necessary to clearly distinguish this population from the 12 study participants who did not receive “early” ART.
2. The authors presented their data on “early” vs. “chronic” treated in separate figures and result sections. There are a lot of individual graphs, different colors for study participants, and many p values. It would be better if the authors found a way to integrate the two sets of data in a manner that is easier to understand for the readers of the journal.
3. The novel finding of this study is that the early initiation of ART allows better virologic control (that is the size of the HIV reservoir) despite a relatively lengthy period of treatment interruption. As the authors speculated, it is very likely that the host immunity against the virus had a lot to do with this phenomenon. Unfortunately, the authors did not conduct any in-depth immune analyses except the measurements of CD4 and CD8 counts. The authors should have pursued relatively simple experiments, such as measurements of the frequency of HIV-specific T cells and examination of immune markers (flow cytometry or biomarkers in plasma), to shed light on the potential mechanism as to how the early treated groups better restricted HIV reservoirs and achieved a favorable immune profile. These experiments do not require a lot of cells and may not result in any useful information. However, such experiments should have been included in this manuscript.
4. It is possible that the authors had a limited number of cells per time point and hence decided to stick with very basic virologic assays (total HIV DNA and cell-associated HIV RNA) to determine the reservoir size. The intact HIV proviral DNA assay (IPDA) is by no means perfect; however, it provides substantially more information concerning the intactness of HIV DNA, and therefore the authors should have included this assay in their manuscript.
5. The authors stated that a lengthy period of treatment interruption has no detrimental impact on immune and virologic markers. The majority of clinicians and researchers in the field would agree with the authors. However, this notion is based on the fact that almost all PWH who restart ART following treatment interruption re-suppress their plasma viremia and recover their CD4 counts not because their viral reservoir size normalizes. In fact, there is no clear evidence of how the HIV reservoir size affects the kinetics of plasma viral rebound. The authors are advised to discuss this issue in the discussion section of the manuscript.
6. It appears that there is no mention of plasma viremia during the treatment interruption period.

Version 1:

Reviewer comments:

Reviewer #1

(Remarks to the Author)

The revised version of the manuscript fully addresses all of my comments and suggestions. The authors did an excellent job creating new figures and modifying some of the paragraphs of the Results and Discussion.

Reviewer #2

(Remarks to the Author)

The authors have sufficiently addressed the concerns I previously raised.

Responses to Reviewers' comments

Reviewer #1:

In the study by Pasternak et al, the authors analyzed the long-term suppressive effect of temporary early ART on the persistence of the viral reservoir. The group had access to a group of unique samples derived from a randomized controlled trial in which they compared 24 or 60 weeks of temporary ART vs. no treatment during primary HIV infection (PHI) with data points longitudinally measured after (re) initiation of ART during chronic HIV infection (CHI) after a median period of ~2.5 years without treatment. The first part of the study, in which the authors compared parameters measured at early ART with the counterparts quantified during chronic ART is not novel and it confirms previous results demonstrating lower HIV-1 reservoir levels and better immune profiles if patients are treated with ART early (e.g, PMID: 34841369, PMID: 33105471, etc.).

The novelty of this study stems from the findings that, in pretreated participants, they observed lower levels of viral persistence markers (cell-associated HIV unspliced RNA and total HIV-1 DNA) during the chronic reinitiation phase compared to those subjects that were not treated early. Interestingly, despite having a long treatment interruption, the authors report that they could not find any significant difference in viral persistence parameters between early and chronic ART; this suggests that it may be relatively safe to withdraw ART during future cure interventions.

We thank the Reviewer for their positive assessment of our study.

The main limitation of the manuscript is the confounding effect of different ART regimens administered at the beginning and end of both early and chronic ART phases. Even though the study design likely does not have enough power to discriminate the effect of different ART regimens on viral persistence (usRNA and HIV DNA) or immune responses (CD4 and CD4/CD8 ratio), this potential confounder must be addressed. Tables 1 and 2 reveal that different regimens were administered to the study subjects at different time points. For example, during CHI, some patients but not all received integrase strand transfer inhibitor-based regimens. The authors merely mentioned (lines 146-147), "neither NRTI backbone nor ART class was significantly different between the study arms (data not shown)". Because the switch to certain ART drugs like dolutegravir has been shown to increase the size of the circulation reservoir (e.g, PMID: 38485563), it should be important to stratify patients during both early and chronic ART based on regimen and determine both parameters of viral persistence and immune reconstitution.

We thank the Reviewer for raising this important point. We now performed this analysis during both early and CHI ART (Figures S4 and S7, respectively). We stratified the participants according to the ART class or NRTI backbone that they received at every time point during ART and compared CD4+ count, CD4/CD8 ratio, US RNA, total DNA, and plasma viral load between participants on different regimens.

During the early ART, no differences for any marker were observed between different NRTI backbones. Concerning the ART class, plasma viral load and total HIV-1 DNA levels were higher at 12 weeks ART in the participants treated with the triple-class, four-drug ART regimen that included a NNRTI combined with a PI, compared to those treated with the three-drug NNRTI-based regimen (Fig. S4A). The reason for this difference is that, by the study protocol, almost all participants started the early ART on the four-drug regimen (NNRTI+PI) and were advised to discontinue the fourth drug when their plasma viral load became undetectable. Consequently, at 12 weeks ART, most of participants whose plasma viral loads were not yet suppressed (and who, therefore, had higher total HIV-1 DNA levels) were still on the

four-drug regimen, while most of those with suppressed viral loads have already discontinued the fourth drug. Therefore, in this case, higher plasma viral load (and total HIV-1 DNA) at 12 weeks is the cause, and not the consequence, of receiving a NNRTI+PI-based (as compared with a NNRTI-based) regimen. No other differences by the ART class were observed for any marker during early ART. This is presented in the revised manuscript, lines 127-141.

During the CHI ART, we only stratified the participants based on the ART class, as the overwhelming majority were receiving tenofovir/emtricitabine as an NRTI backbone. Concerning the ART class, most participants were receiving NNRTI or a ritonavir-boosted PI as a third drug, although a minority were indeed receiving an INSTI-based regimen (Fig. S7). INSTI-based regimen was not associated with higher levels of reservoir markers as compared with NNRTI- or PI-based regimens. In fact, no differences by the ART class were observed for any marker (Fig. S7A), and neither NRTI backbone nor ART class was significantly different between the study arms during CHI ART (Fig. S7B). This is presented in the revised manuscript, lines 164-169.

A minor limitation is the interpretation of the transcriptional activity of the HIV-1 reservoir. On line 108, the authors claimed that it "remained stable afterward". However, in Figure 2A, they showed statistically significant differences in the levels of US RNA when they compared 24w. vs. 36w and 36w. vs. 48w. They should rephrase these findings.

The statement that US RNA remained stable after week 12 is based on the lack of a significant trend in time after that time point (Fig. 3A). However, some significant fluctuations in the levels of US RNA were indeed observed. We rephrased this sentence to refer to these fluctuations (lines 113-114).

Reviewer #2:

Pasternak et al. conducted limited immune and virologic analyses on clinical specimens collected from a previous/follow-up clinical study addressing the impact of early initiation of antiretroviral therapy (ART) on persistent HIV reservoirs and CD4/CD8 counts in people with HIV (PWH). The major findings of this study include: 1) the size of HIV reservoirs is lower in PWH who initiated ART during the early phase of infection, 2) the reservoir size in the early ART group remains relatively low in early treated populations despite having undergone a relatively lengthy treatment interruption, and 3) this is one of the first longitudinal studies addressing the effect of short-term ART in PWH who later became chronic-infected and subsequently reinitiated ART. Although the data presented in this manuscript are somewhat expected, this study is unique in that the key measurements were done in the longitudinal early-treated study cohort who underwent treatment interruption and ART. However, the current manuscript lacks any explanation regarding the underlying mechanism(s) by which the early initiation of ART allows better immunologic and virologic control in the study populations examined.

We thank the Reviewer for their positive assessment of our study. We have now measured a number of additional immunological markers in plasma and PBMCs in order to provide an explanation for the underlying mechanism (see below).

1. This manuscript, especially the abstract section, could have benefited tremendously if the authors had clearly defined the study populations and the time points when the assays were conducted. Unless you are intimately familiar with the original study design and read the rest of the manuscript first, it is challenging to comprehend the abstract mainly because the "early" population eventually became "chronic" during treatment interruption and it is not clear what time points immune and virologic markers were measured (i.e., during initial or later ART period). It may not be a bad idea to give each

population an acronym (for example, “Early 24” etc). In addition, given the early study participants eventually became chronic before reinitiation of ART, it is necessary to clearly distinguish this population from the 12 study participants who did not receive “early” ART.

We have made an effort to clarify the study design. We amended the abstract to make it clearer at which periods which study arms were assessed. We also modified Figure 1 to include acronyms suggested by the Reviewer.

2. The authors presented their data on “early” vs. “chronic” treated in separate figures and result sections. There are a lot of individual graphs, different colors for study participants, and many p values. It would be better if the authors found a way to integrate the two sets of data in a manner that is easier to understand for the readers of the journal.

We have modified Figures 2 and 3 to make the data presentation more straightforward. We now integrated the levels of the parameters during both early and CHI ART in one figure (new Figure 2) and their individual longitudinal trajectories, also during both early and CHI ART, in another figure (new Figure 3). We also removed most of the p values from Figure 2 and show them in the new supplementary figure (Figure S2).

3. The novel finding of this study is that the early initiation of ART allows better virologic control (that is the size of the HIV reservoir) despite a relatively lengthy period of treatment interruption. As the authors speculated, it is very likely that the host immunity against the virus had a lot to do with this phenomenon. Unfortunately, the authors did not conduct any in-depth immune analyses except the measurements of CD4 and CD8 counts. The authors should have pursued relatively simple experiments, such as measurements of the frequency of HIV-specific T cells and examination of immune markers (flow cytometry or biomarkers in plasma), to shed light on the potential mechanism as to how the early treated groups better restricted HIV reservoirs and achieved a favorable immune profile. These experiments do not require a lot of cells and may not result in any useful information. However, such experiments should have been included in this manuscript.

We thank the Reviewer for this important suggestion. To address this point, we measured a number of immunological markers in plasma and PBMCs, during both early and CHI ART.

First, we measured functional HIV-specific T-cell responses: CD4+ and CD8+ T-cell reactivity (by the activation-induced marker (AIM) assay), IFN- γ release, as well as CD4+ and CD8+ T-cell proliferative responses, upon stimulation with Gag peptide pools. We compared these responses between early and CHI ART in the same participants (Fig. 5D) and also during CHI ART between the study arms (Fig. S14B). Robust CD4+ and CD8+ T-cell responses were observed at both early and CHI ART. Gag-specific CD8+ T-cell reactivity was significantly higher during CHI ART as compared to the early ART in the same participants, accompanied by a nonsignificant trend towards higher IFN- γ release during CHI ART (Fig. 5D). During the CHI ART, no significant differences were observed between the arms for any of the measured HIV-specific T-cell response parameters (Fig. S14B). This is presented in the revised manuscript, lines 247-256 and 334-344.

Second, we measured plasma soluble biomarkers of systemic inflammation (IL-6), intestinal damage (I-FABP), and monocyte activation (sCD163 and sCD14). Interestingly, levels of IL-6 and I-FABP were significantly higher, while both monocyte activation markers were significantly lower, on early ART compared to CHI ART in the same participants (Fig. 5E). During the CHI ART, no significant differences between the arms were observed for any of these markers as well (Fig. S14C). This is presented in the revised manuscript, lines 257-262 and 344-347.

4. It is possible that the authors had a limited number of cells per time point and hence decided to stick with very basic virologic assays (total HIV DNA and cell-associated HIV RNA) to determine the reservoir size. The intact HIV proviral DNA assay (IPDA) is by no means perfect; however, it provides substantially more information concerning the intactness of HIV DNA, and therefore the authors should have included this assay in their manuscript.

To address this important point, we performed IPDA to compare levels of intact and defective proviruses in participants with paired samples obtained at early and CHI ART. Similarly to what was observed for US RNA and total DNA, no significant difference in the levels of intact proviruses between the therapy periods was measured in the same participants (Fig. 6D). However, significantly higher levels of total defective (3' defective + 5' defective) proviruses were observed at CHI ART (Fig. 6D). In spite of this difference, total (intact + defective) proviruses did not significantly differ between the two therapy periods (Fig. 6D). We also observed strong positive correlations between the two therapy periods for all proviral forms measured: intact, 3' defective, and 5' defective proviruses (Fig. 6D). These results strongly suggest that, upon reinitiation of ART during CHI, not only US RNA and total DNA but also intact HIV-1 reservoirs returned to their early-ART levels despite the long TI. At the same time, we observed an increase in defective proviruses at CHI ART compared to early ART, which may be explained by clonal expansion of cells harbouring defective proviruses. This is presented in the revised manuscript, lines 286-302.

We also determined correlations between the HIV-1 proviral forms and HIV-specific T-cell responses at both early and CHI ART. At early ART, intact HIV-1 DNA weakly negatively correlated with CD8+ T-cell reactivity and IFN- γ release but a correlation of 5' defective HIV-1 DNA with IFN- γ release was stronger (Fig. S12). However, at CHI ART, intact HIV-1 DNA significantly negatively correlated with CD8+ T-cell proliferation, reactivity, and IFN- γ release (Fig. 6E), while correlations of defective HIV-1 DNA with T-cell responses were weaker (Fig. S12). This is presented in the revised manuscript, lines 303-310.

These results suggest that HIV-specific T-cell responses indeed restrict the intact HIV-1 reservoir during ART, especially during ART initiated at CHI.

5. The authors stated that a lengthy period of treatment interruption has no detrimental impact on immune and virologic markers. The majority of clinicians and researchers in the field would agree with the authors. However, this notion is based on the fact that almost all PWH who restart ART following treatment interruption re-suppress their plasma viremia and recover their CD4 counts, not because their viral reservoir size normalizes. In fact, there is no clear evidence of how the HIV reservoir size affects the kinetics of plasma viral rebound. The authors are advised to discuss this issue in the discussion section of the manuscript.

We added this interesting point to the Discussion (lines 379-384). We also refer to a number of studies showing that HIV-1 US RNA, measured under ART, does predict viral rebound upon TI (lines 401-403). These studies include our previous publication on the Primo-SHM cohort (PMID: 32097124), where we demonstrated that US RNA measured just before TI, predicted both time to and magnitude of the plasma viral load rebound following the TI.

6. It appears that there is no mention of plasma viremia during the treatment interruption period.

The levels of plasma viremia during CHI (at the baseline of CHI ART) are shown in Figure 2B. Also, the levels of plasma viremia at the post-TI virological setpoint were reported in our previous publication

(PMID: 32097124, supplemental Figure 1), as well as in the first publication on the Primo-SHM cohort (PMID: 22479156, Figure 2).